# Synergic Effect of Robot-Assisted Rehabilitation and Antispasticity Therapy: A Narrative Review

**DOI:** 10.3390/life13020252

**Published:** 2023-01-17

**Authors:** Wei-Cheng Wang, Chia-Yi Yeh, Jian-Jia Huang, Shih-Chieh Chang, Yu-Cheng Pei

**Affiliations:** 1Department of Medical Education, Chang Gung Memorial Hospital c, 5 Fushing St., Taoyuan 333, Taiwan; 2Department of Orthopedic Surgery, Chang Gung Memorial Hospital at Linkou, 5 Fushing St., Taoyuan 333, Taiwan; 3Department of Physical Medicine and Rehabilitation, Chang Gung Memorial Hospital at Linkou, 5 Fushing St., Taoyuan 333, Taiwan; 4Graduate School of Science Design Program in Innovation for Smart Medicine, Chang Gung University, 259 Wen-Hwa 1st Road, Taoyuan 333, Taiwan; 5Center of Vascularized Tissue Allograft, Chang Gung Memorial Hospital at Linkou, No. 5 Fushing St., Taoyuan 333, Taiwan; 6School of Medicine, Chang Gung University, 259 Wen-Hwa 1st Road, Taoyuan 333, Taiwan

**Keywords:** stroke, spinal cord injury, robotic-assisted therapy, antispasticity therapy, rehabilitation

## Abstract

Background: Stroke and spinal cord injury are neurological disorders that cause disability and exert tremendous social and economic effects. Robot-assisted training (RAT), which may reduce spasticity, is widely applied in neurorehabilitation. The combined effects of RAT and antispasticity therapies, such as botulinum toxin A injection therapy, on functional recovery remain unclear. This review evaluated the effects of combined therapy on functional recovery and spasticity reduction. Materials and Methods: Studies evaluating the efficacy of RAT and antispasticity therapy in promoting functional recovery and reducing spasticity were systemically reviewed. Five randomized controlled trials (RCTs) were included. The modified Jadad scale was applied for quality assessment. Functional assessments, such as the Berg Balance Scale, were used to measure the primary outcome. Spasticity assessments, such as the modified Ashworth Scale, were used to measure the secondary outcome. Results: Combined therapy improves functional recovery in the lower limbs but does not reduce spasticity in the upper or lower limbs. Conclusions: The evidence supports that combined therapy improves lower limb function but does not reduce spasticity. The considerable risk of bias among the included studies and the enrolled patients who did not receive interventions within the golden period of intervention are two major factors that should be considered when interpreting these results. Additional high-quality RCTs are required.

## 1. Introduction

Stroke and spinal cord injury (SCI) are common diagnoses among patients receiving inpatient rehabilitation and often limit such patients’ abilities to perform activities of daily living. Most patients with stroke or SCI experience functional impairment, spasticity, and motor clumsiness [1], and spasticity and functional impairment are the primary reason for rehabilitative intervention in the chronic stages of recovery [2,3,4]. A major line of inquiry in rehabilitation sciences is the development of therapies optimized for managing spasticity and functional impairment in such patients.

Spasticity is a motor disorder that manifests as an increase in tonic stretch reflexes caused by hyperexcitability of the stretch reflex in patients with upper motor neuron disorders, such as stroke and SCI [5]. Spasticity occurs in 4–27% and 17–46% of patients with stroke within the first month and first 3 months after stroke, respectively, and in 70% of patients with SCI [6,7,8,9]. Spasticity limits the performance of daily activities, social participation, and quality of life [10]. Controlling spasticity is essential to alleviating its resulting functional impacts [11,12]. Therefore, a major line of inquiry in rehabilitation sciences is the development of therapies optimized for managing spasticity.

In recent 10 years, robot-assisted training (RAT) has been applied for improving impaired motor function in patients with neurological disorders. RAT is superior to conventional therapies for neurorehabilitation in both the upper [13,14] and lower limbs [15] because it integrates high-intensity, repetitive, and task-specific training. It can also measure a patient’s performance with high reliability and accuracy [16]. However, despite the evident utility of RAT, whether a combination of RAT and other therapies can yield a synergic effect remains unclear.

Studies have established the effectiveness of RAT and antispasticity therapy for patients with upper motor neuron diseases. Some researchers have suggested that combining these two types of therapy may result in further improvement of functional outcomes or reductions in spasticity effects. A previous systematic review reported that botulinum toxin A (BoNT) injection therapy alone may improve motor function. However, a combination of BoNT injection and rehabilitation exerted a limited synergic effect on motor functional recovery [17]. Furthermore, the effect of combined therapy on spasticity and the effects of antispasticity therapies apart from BoNT have not yet been thoroughly investigated. To this end, the present narrative review was conducted to assess whether a combination of RAT and antispasticity therapy exerts a synergic effect on functional recovery and spasticity in patients with upper motor neuron disorders.

## 2. Materials and Methods

### 2.1. Inclusion Criteria

#### 2.1.1. Types of Studies

Only randomized controlled trials (RCTs) were included in the review; no publication status restrictions were applied. We included both blinded and nonblinded studies because blinding a patient or therapist to whether RAT is being administered is impossible. We excluded midterm reports of ongoing studies and midterm reports related to the included RCTs. The searches were not limited to English literature; publications in other languages were examined using their English or translated abstracts.

#### 2.1.2. Types of Participants

We included studies that employed samples comprising patients 18 years or older with any upper motor neuron disease of any etiology and disease severity, regardless of their age, gender, ethnicity, and native language. Among five trials included in our study, four involved patients with stroke, and the remaining one involved patients with SCI.

#### 2.1.3. Types of Interventions

We included trials that investigated the effects of RAT combined with antispasticity therapy. In the studies that investigated the additive effect of RAT, the control group received no RAT; in the studies that investigated the additive effect of antispasticity therapy, the control group received no antispasticity therapy. We excluded studies in which robotic devices were used only as evaluation tools.

#### 2.1.4. Types of Outcome Measures

We included trials in which the outcomes were assessed in terms of motor function, spasticity, or both. The primary outcome was functional recovery, which was measured using the Fugl–Meyer Assessment (FMA), Medical Research Council Scale (MRC), and Box and Block Test (B&B) for the upper extremities and the 6 min walk test (6MWT), timed up-and-go test [18], Berg Balance Scale (BBS), Rivermead Visual Gait Assessment (RVGA), and 10 m walk test (10MWT) for the lower extremities. The secondary outcome was spasticity, which was measured using the modified Ashworth Scale (MAS) and Tardieu Scale.

### 2.2. Search Methods for Identification of Studies

We performed a systematic online literature search of the PubMed, Cochrane Library, Embase, and Scopus databases from inception to 31 July 2022, using the following medical subject heading terms and search strategy: Search 1: spasticity OR spastic OR “upper motor neuron syndrome”; Search 2: robot OR robotic; Search 3: baclofen OR clonidine OR tizanidine OR benzodiazepines OR diazepam OR clonazepam OR gabapentin OR dantrolene OR botulinum OR Botox OR “phenol injection” OR “alcohol injection” OR “intrathecal baclofen” OR neurolysis OR “nerve block” OR “dorsal rhizotomy” OR neurectomy; Final Search: Search 1 AND Search 2 AND Search 3. In addition to the aforementioned database searches, the references of identified articles were examined.

### 2.3. Data Collection and Analysis

#### 2.3.1. Selection of Studies

Two authors (WCW and CYY) independently screened the titles and abstracts of the studies obtained through the search by using our inclusion and exclusion criteria. Thereafter, the full texts of the 190 papers that passed the initial screening were examined. At each stage, a third reviewer was consulted for discrepancies between reviewers, and a consensus was reached in all cases.

#### 2.3.2. Data Extraction and Management

The country of origin, participant characteristics (limbs examined, disease, mean age, and gender), trial setting, sample size, intervention details, and outcome measures were extracted from the five included studies by two authors. Discrepancies between the authors were resolved through discussion.

#### 2.3.3. Methodological Quality Assessment

We evaluated the methodological quality of the included studies by using the modified Jadad scale. A detailed explanation of the quality assessment is provided in Table 1. Because the goal of this review was to analyze all studies evaluating the clinical efficacy of combined therapy for neurological disorders, no study was excluded on the basis of quality.

#### 2.3.4. Assessment of Reporting Biases

We conducted a comprehensive search to minimize reporting bias. Because fewer than 10 trials were included, we were unable to evaluate potential publication bias using the funnel plot.

#### 2.3.5. Qualitative Analysis

Because the patient populations, interventions, and outcome measures of the included studies varied widely, we refrained from statistical pooling. We therefore performed qualitative synthesis of the data for the outcomes of interest. Results were considered contradictory if the overall conclusions regarding the effectiveness of the same intervention for the same indication in different studies were contradictory or inconsistent.

#### 2.3.6. Reaching Conclusions

Our conclusions were drawn according to the findings from the qualitative synthesis of the included studies. We developed recommendations and determined the implications of our results for practice and future research accordingly.

## 3. Results

### 3.1. Literature Search

A total of 190 studies were identified in the initial database search. After the titles and abstracts were evaluated, the full texts of nine of the studies were assessed for eligibility. Ultimately, five studies [19,20,21,22] were included in the narrative review. Figure 1 illustrates the literature search and the reasons for exclusion.

### 3.2. Risk-of-Bias Assessment

The risk of bias in the included studies was assessed using the modified Jadad scale (Table 1). The scores of the included studies ranged from 3 to 5.5, indicating a considerable risk of bias. The methods of randomization and blinding were the major causes of the high risk of bias.

### 3.3. Features of Included Studies

Table 2 summarizes the characteristics of the included studies. All the eligible studies were RCTs published between 2015 and 2019. Four of the studies enrolled patients with stroke, and one enrolled patients with SCI. Two of the studies (Gandolfi et al., 2019 [19] and Pennati et al., 2015 [20]), both of which enrolled patients with stroke, assessed motor function of the upper limbs. The other three studies, among which two (Picelli et al., 2016 [21] and Erbil et al., 2018 [18]) enrolled patients with stroke and one (Duffell et al., 2015 [22]) enrolled patients with SCI, assessed the motor function of the lower limbs. Each of the included studies enrolled 15 to 48 patients, and 29% to 40% of each sample consisted of female patients. The mean ages of the patients ranged from 46.5 to 65.1 years. The mean intervals between disease onset and enrollment ranged from 6 months to 20 years. The number of RAT sessions ranged from 10 to 15, and the intervention durations ranged from 5 days to 5 weeks. The mean follow-up intervals ranged from 0 days (immediately after treatment) to 12 weeks.

### 3.4. Intervention Protocols

The intervention protocols employed in the included studies varied widely (Table 3). Two of the studies involved upper limb interventions: that by Gandolfi et al., who recruited 32 patients and compared BoNT + RAT with BoNT + physical therapy, and that by Pennati et al., who recruited 15 patients and compared BoNT + RAT with RAT alone. The studies by Picelli et al., who recruited 22 patients and compared BoNT + robot-assisted gait training (RAGT) with BoNT alone; Erbil et al., who recruited 43 patients and compared BoNT + RAGT + PT with BoNT + PT; and Duffell et al., who recruited 48 patients and compared RAGT + oral tizanidine with RAGT alone, all focused on lower limb interventions.

### 3.5. Primary Outcome: Functional Recovery

The primary objective of our narrative review is to determine whether the patients who received combined therapy achieved superior functional outcomes (Table 3). In the two studies, the evaluated functional improvement in the upper limbs, the FMA, MRC, and B&B were used to measure the outcomes (Table 3). Pennati et al. reported that the FMA and B&B scores of the patients who received RAT alone improved significantly more than did those of the patients who received combined therapy (BoNT + RAT), whereas Gandolfi et al. reported no significant differences in FMA and MRC score improvements between the patients who received BoNT + PT and those who received BoNT + RAT. In summary, the included studies yielded conflicting conclusions regarding the effect of combination therapy on functional recovery in the upper limbs.

In the three studies that assessed functional improvement in the lower limbs, the outcomes were measured using the 6MWT, BBS, TUG, RVGA, and 10MWT (Table 3). Picelli et al. reported that the 6MWT results of the patients who received combined therapy (BoNT + RAT) improved more than those of the patients who received BoNT alone. Erbil et al. reported that the BBS, TUG, and RVGA scores of the patients who received combined therapy (BoNT + PT + RAGT) improved more than those of the patients who received BoNT + PT. Duffell et al. reported that the 10MWT results of the patients who received RAGT + oral tizanidine improved discernably (according to the minimal important difference), whereas those of the patients who received RAGT alone did not. In summary, combined therapies are more effective in promoting functional recovery in the lower limbs.

Overall, our findings suggest that combined therapy improves functional recovery in the lower limbs; however, no evidence supports its therapeutic effect on upper limb function.

### 3.6. Secondary Outcome: Spasticity

Our secondary outcome was spasticity reduction (Table 3). To determine whether the patients who received combined therapy experienced greater reductions in spasticity than those who received other interventions, we analyzed the spasticity assessment results in the included reports. Four of the studies used the MAS, two used the Tardieu Scale, and one (Duffell et al.) did not report spasticity. The two studies that assessed spasticity by using the MAS focused on the upper limbs. Pennati et al. reported that the MAS scores of the patients who received combined therapy (BoNT + RAT) decreased more than those of the controls; however, Gandolfi et al. did not report similar results.

Two of the studies that focused on the lower limbs (Picelli et al. and Erbil et al.) adopted the MAS and Tardieu Scale to evaluate spasticity. In these two studies, the scores of the MAS or Tardieu Scales of the patients in the combined therapy groups did not decrease more than those of the controls. Overall, our findings indicate that combined therapy does not reduce spasticity in the upper limbs or lower limbs, except in the upper limbers of patients receiving BoNT + RAT.

### 3.7. Power Analysis and Effect Size

Only two of the included studies reported effect sizes: that by Gandolfi et al., in which the effect size ranged from −0.02 to 0.49, and that by Picelli et al., in which the effect size ranged from 0.07 to 0.47. However, the methods employed for effect size estimation in these studies were not delineated. These studies also performed power analysis, which revealed that their sample sizes were sufficient for avoiding type II errors. Because the other three studies did not report their effect sizes and did not perform power analysis, the effectiveness of the interventions evaluated therein could not be adequately inferred, and their results could have been a result of inadequate sample size.

## 4. Discussion

This is the first narrative review to compare the effectiveness of the combination of RAT and antispasticity therapy with that of antispasticity therapy or RAT alone. The results suggest that combined therapy can promote functional improvement in the lower limbs, but not in the upper limbs. Furthermore, combined therapy, such as BoNT + RAT, BoNT + RAGT, BoNT +PT + RAGT, or Tizanidine + RAGT, does not seem to reduce spasticity in the upper or lower limbs. However, because the patients in the included studies were mainly recruited over 6 months after disease onset, the results of the present review may only be applicable to patients in chronic stages of recovery after stroke or SCI. Furthermore, we identified a high risk of bias among the included studies; thus, these results may not be representative of the true effects of interventions. Nevertheless, we suggest applying combined therapy to improve functional recovery in the lower limbs of patients with chronic stroke or SCI.

An important issue is the possible mechanisms underlying the finding that RAT improves functional recovery in the lower but not in the upper limbs. This difference could be reflected by the difference in the movement capability between upper- and lower-limb robots. The movement performed by upper limbs, especially the hand, is more complicated as hand movements, such as grasp or pinch, are mediated by a combination of multiple small-joint movements, including interphalangeal, metacarpophalangeal, intercarpal, radiocarpal, and radioulnar joint [23,24]. Compared to the movement of lower limbs, these upper-limb movements cannot be fully supported by upper-limb robots [25,26,27,28,29], therefore the current upper limb RAT is less effective as compared with lower limb RAT.

RAT integrates high-intensity, repetitive, and task-specific exercises into the treatment of patients with functional impairment. Studies have reported that RAT is as effective as conventional training for improving upper and lower limb motor function, facilitating activities of daily living, and reducing spasticity [30]. The present study investigated whether a combination of BoNT injection therapy and RAT could exert a synergic effect on functional outcomes and spasticity.

The management of spasticity is crucial when applying RAT to patients with upper motor neuron disorders. Several patient-related factors, including spasticity, may interfere with RAT. For instance, spasticity limits a patient’s range of motion and functional recovery in the upper and lower limbs [10,31]. BoNT injection therapy is widely applied and considered a safe and effective treatment for managing spasticity [32,33]. Furthermore, patients who receive BoNT injection therapy after stroke experience improvements in functional outcomes and reduced spasticity [34].

The differences in the interventions and outcome assessments employed in the included studies resulted in high heterogeneity in our sample. First, the intervention methods, including the selection of robotic devices and training protocols and the dosage of BoNT injections, differed among the included studies. Specifically, two of the studies applied end effectors for upper limb training, one applied end effectors for lower limb training, and two applied exoskeleton devices for lower limb training. A previous study reported that patients who underwent gait training using end effectors had a higher rate of independent walking than those who underwent gait training using exoskeleton devices [35]. Therefore, the standardization of RAT is crucial to yield more consistent results in future studies. Second, the characteristics of different upper motor neuron disorders, such as stroke and SCI, are distinct. In addition to impairments in motor function, patients with stroke are prone to executive function deficits and thus require additional environmental and procedural accommodations during intervention [36]. Therefore, studies involving patients with stroke should comprehensively evaluate the cognitive status and motivation of the recruited patients. Third, two of the studies included in the present review evaluated spasticity and functional recovery in the upper limbs, whereas three focused on the lower limbs. In patients with stroke, the patterns of recovery in the upper and lower limbs differ: functional recovery of the upper limbs occurs later but is more persistent than that of the lower limbs [37]. Therefore, the design of follow-up assessments is crucial for the accurate documentation of changes in functional outcomes over time. Fourth, due to the limited subjects in each study, the characteristics between the intervention group and control group are not equally distributed. Specifically, the result of Pennati et al. showed that the FMA and B&B scores of the patients who received RAT alone improved significantly more than those of the patients who received combined therapy (BoNT + RAT). This finding could be accounted for by the fact that pre-treatment functional impairment of patients in the combined therapy group was more severe than the RAT alone group. In summary, because of the high heterogeneity among the included studies, which is attributable to the various intervention protocols employed, the unique characteristics of upper motor neuron disorders, and the distinct patterns of recovery in the upper and lower limbs, we were unable to thoroughly integrate the results of the included studies to obtain a precise conclusion regarding the efficacy of combined therapy.

Several characteristics of our literature sample, including delayed interventions, limited case numbers, and heterogeneity in study design, limit the ability of our review to draw a firm conclusion regarding the clinical efficacy of combined therapy. First, because the intervals between neurological injury and patient enrollment were over 6 months in most of the included studies, the enrolled patients did not receive interventions within the golden period of intervention, which is defined as the first 3 to 6 months after neurological injury [38,39]. Because patients who initiate neurorehabilitation after this period are less likely to exhibit considerable functional improvement, evaluating the overall effectiveness of interventions is difficult. Second, all the included studies had small sample sizes, indicating that most of the researchers did not conduct power analysis before initiating their respective studies. Small sample sizes and low effect sizes could result in false negatives. Although meta-analysis could overcome this limitation by integrating the results of multiple studies, the high heterogeneity among the outcome measures used in the included studies eliminated this possibility. Third, because most of the included studies did not report the randomization method, blinding, or adverse effect assessments used, our sample exhibited a considerable risk of bias, and caution must be taken when interpreting our results.

## 5. Conclusions

The results of our narrative review suggest that combined therapy may improve lower limb function. However, our results must be interpreted with caution because of the considerable risk of bias among the included studies and because the enrolled patients were not within the golden period of intervention. Regarding the other outcomes of interest, namely functional recovery of the upper limbs and spasticity reduction in the upper and lower limbs, the findings are controversial, and additional high-quality RCTs are required.

To improve the overall quality of the available evidence, future investigations may adopt several measures. First, the selection of robotic devices and training protocols in RAT should be standardized to ensure that the outcome measurements of future studies are comparable on the same basis. Second, outcome measures employed in future studies should be more closely related to RAT to ensure accurate evaluation of the clinical efficacy of RAT. Third, future studies should employ high-quality study designs, including effective randomization methods and blinding, to yield high-quality evidence. Finally, future studies should enroll patients who are within the golden period of intervention and who therefore exhibit greater potential for functional recovery.

## Figures and Tables

**Figure 1 life-13-00252-f001:**
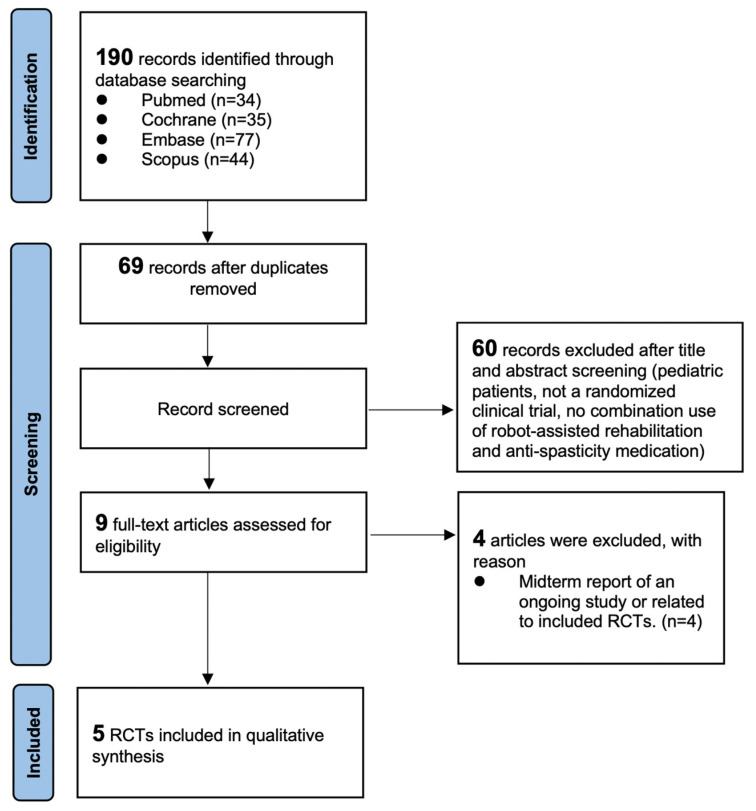
PRISMA flow diagram of study inclusion.

**Table 1 life-13-00252-t001:** Methodological quality assessment of included studies (modified Jaded scale).

Question	Gandolfi et al., 2019 [19]	Pennati et al., 2015 [20]	Picelli et al., 2016 [21]	Erbil et al., 2018 [18]	Duffell et al., 2015 [22]
Was the study described as randomized?	Yes	Yes	Yes	Yes	Yes
Was the method of randomization appropriate?	Yes	Yes	Yes	Not described	Not described
Was the study described as blinding?	Single blind	Single blind	Single blind	No	No
Was the method of blinding appropriate?	No	No	No	No	No
Was there a description of withdrawals and dropouts?	Yes	Yes	Yes	Yes	Yes
Was there a clear description of the inclusion/exclusion criteria?	Yes	Yes	Yes	Yes	Yes
Was the method used to assess adverse effects described?	No	No	Yes	No	No
Was the methods of statistical analysis described?	Yes	No	Yes	Yes	Yes
Score	4.5	3.5	5.5	3	3

**Table 2 life-13-00252-t002:** Demographic data of patients in included studies.

Author, Year	Disease	Limbs of Evaluation	Total No. of Patients (F/M)	Group	No. of Patients (F/M)	Mean Age, Year (SD)	Time since Injury (SD)
Gandolfi et al., 2019 [19]	Stroke	UL	32 (10/22)	Control	16 (6/10)	59.13 (14.97)	5.1 yr (2.2 yr)
				Intervention	16 (4/12)	59.31 (14.40)	6.0 yr (3.1 yr)
Pennati et al., 2015 [20]	Stroke	UL	15 (6/9)	Control	8 (NA)	NA	>6 mo, (10 mo to 20 yr)
				Intervention	7 (NA)	NA	>6 mo, (10 mo to 20 yr)
Picelli et al., 2016 [21]	Stroke	LL	22 (6/16)	Control	11 (4/7)	65.1 (3.4)	6.1 yr (3.8 yr)
				Intervention	11 (2/9)	62.4 (9.5)	6.2 yr (4.2 yr)
Erbil et al., 2018 [18]	Stroke	LL	43 (16/27)	Control	14 (3/11)	48.7 (10.4)	25.9 mo (24.6 mo)
				Intervention	29 (13/16)	50.1 (11.8)	39 mo (34.3 mo)
Duffell et al., 2015 [22]	SCI	LL	48 (14/34)	Control	26 (7/19)	46.6 (12.6)	9.3 yr (8.9 yr)
				Intervention	22 (7/15)	46.5 (11.9)	10.2 yr (10.47 yr)

SCI: spinal cord injury; UL: upper limbs; LL: lower limbs; NA: not available; F: female; M: male; mo: month; yr: year.

**Table 3 life-13-00252-t003:** Interventions and outcome measurements of included studies.

Author, Year	Group	Intervention Details	Functional Measurement	Spasticity Measurement	Follow Up
Gandolfi et al., 2019 [19]	Control	BoNT + PT (Individualized dosage and targe muscle of BoNT) (Mobilization and stretching (10 min) followed by UL exercises of PT, 10 sessions within 5 weeks)	FMA *, MRC ^§^	MAS *	Post-treatment
	Intervention	BoNT + RAT (BoNT as above) (End-effector device of RAT with Armotion [Reha Technology, Olten, Switzerland], 10 sessions within 5 weeks)	FMA *, MRC *^§^	MAS *	
Pennati et al., 2015 [20]	Control	RAT (End-effector device of RAT with ReoGo System [Mo-torika Medical Ltd.; Caesarea, Israel], 10 sessions within 3–5 weeks)	FMA *^§^, B&B *^§^	MAS *^§^	End of RAT
	Intervention	BoNT + RAT (Individualized dosage and targe muscle of BoNT) (RAT as above)	FMA *^§^, B&B*^§^	MAS *^§^	
Picelli et al., 2016 [21]	Control	BoNT (GM, GL and soleus muscle, 250U each muscle)	6MWT ^§^	MAS *, Tardieu *	4 weeks
	Intervention	BoNT + RAGT (BoNT as above) (End-effector static robot of RAGT with G-EO System Evolution [Reha Technology, Olten, Switzerland], 5 sessions within 1 week)	6MWT *^§^	MAS *, Tardieu *	
Erbil et al., 2018 [18]	Control	BoNT + PT (Individualized dosage and targe muscle of BoNT) (15 sessions within 3 weeks: 90 min of PT with stretch-ing and strengthening exercises, proprioception, weight bearing, balance, coordination and ambulatory training)	BBS *^§^, TUG *^§^, RVGA *^§^	MAS *, Tardieu *	6, 12 weeks
	Intervention	BoNT + PT + RAGT (BoNT and PT as above) (End-effector static robot of RAGT with RoboGait [Bama Teknoloji, Ankara, Turkey], 15 sessions within 3 weeks: 30 min of RAGT + 60 min of PT)	BBS *^§^, TUG *^§^, RVGA *^§^	MAS *, Tardieu *	
Duffell et al., 2015 [22]	Control	RAGT (Treadmill-base exoskeletal static robot of RAGT with Lokomat [Hocoma AG, Switzerland], 12 sessions within 4 weeks)	6MWT, 10MWT *, TUG	NA	0, 1, 2, 4 weeks (from the start of RAGT)
	Intervention	Tizanidine + RAGT (0.03 mg/kg QID, initiated 4 weeks prior to RAGT) (RAGT at above)	6MWT, 10MWT *^※^, TUG	NA	

* Within-group differences; § between-group differences; ※: minimal important difference; NA: not available; F: female; M: male; BoNT, botulinum toxin type A injection; RAT: robot-assisted training; RAGT, robot-assisted gait training; PT: physical therapy; UL: upper limbs; GM: gastrocnemius medialis; GL: gastrocnemius lateralis; MAS: modified Ashworth Scale; Tardieu: Tardieu Scale; FMA: Fugl–Meyer Assessment; MRC: Medical Research Council Scale; B&B: Box and Block Test; 6MWT: 6 min walk test; BBS: Berg Balance Scale; TUG: timed up-and-go test; RVGA: Rivermead Visual Gait Assessment; 10MWT: 10 m walk test, QID: four times a day.

## Data Availability

The data associated with the paper are available in the following references: Duffell et al., 2015 [23], Erbil et al., 2018 [22], Picelli et al., 2016 [21], Pennati et al., 2015 [20], Gandolfi et al., 2019 [19].

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
