# Peer review of "Synergic Effect of Robot-Assisted Rehabilitation and Antispasticity Therapy: A Narrative Review"

_life, 2023, doi:10.3390/life13020252_

Round 1

Reviewer 1 Report

The objective of this research was conducted to investigate and to assess whether a combination of RAT and antispasticity therapy exerts a synergic effect on functional recovery and spasticity in patients with upper motor neuron disorders.

The approach is original. The manuscript reads smoothly and is easy to understand. The aims, scope, and results of the study are clearly stated. I have very much enjoyed reading this paper. I find it interesting and clearly written and satisfying also all the other publication criteria of the “Life”. The study provides a very valuable addition to this line of research, and adds relevantly to the subject with additional original findings. I thus find that this paper definitively delivers results that will surely be of interest to the readership of the “Life”. Two things bother me about accepting this paper and I would let the editor make the decision:

The first, only 5 RCTs included in the qualitative synthesis, this impacts the conclusion of this paper. The second point is the limits claimed by the authors in the conclusion area showing the strong limit of such as study.

major concerns is that rehabilitation protocols were not described (RAGT exercise typologies) in table as average dose of Botulinum toxins targets or associated physical exercises 

Reviewer 2 Report

The authors have conducted a review of published RCTs on the combination of Robot-Assisted Training (RAT) and combined treatment by either BoNT or Tizanidine versus RAT as a single treatment. The authors found 5 RCTs, four of which based on patients with a combination of stroke and BoNT-treatment and one with patients with SCI and Tizanidine.

The rationale for the study as well as the methods employed are well described, and the authors seem to have performed a thorough analysis of the included studies, and more specifically their limitations (bias, heterogenicity in many aspects and so on, e.g “The differences in the interventions and outcome assessments employed in the included studies resulted in high heterogeneity in our sample”).

Remarks:

1.     Given the fact that there were so few RCT and so heterogeneous, should the literature review not have been expanded to level II and III studies as well? 

2.     Is the conclusion that combined therapy improves functional recovery in the lower but not in the upper limbs really adequate from this limited number of studies, and what would be possible explanations for this? Or could this simply be an indication that the number and quality of level I studies this far is too low to really tell? 

3.     In general, the background on RAT could be more comprehensive. The statement of ”evident utility” at line 72 of RAT has to backed somewhat more by references in my opinion. For instance:  The authors state at Line 68-70 that “RAT is superior to conventional therapies for neurorehabilitation because it integrates high-intensity, repetitive, and task-specific training “, but reference 13 deals only with the upper limb and is not altogether that enthusiastic. The authors of that study found explicitly no evidence that RAT improved the outcome in their patients(“no overall significant effect in favor of robot-assisted therapy was found in the present meta-analysis”). Furthermore: Reference 22 states that “for treatment sustainability of upper and lower limbs, robotic training is just as effective as conventional training”. If conventional training is as effective, why opt for a more complicated regime?

4.     In table 3, the details of the number of sessions for intervention is missing for Pennati and for BONT in Picelli.

5.     At line 197 in the primary outcome section, the authors report that “ Pennati et al. reported that the FMA and B&B scores of the patients who received RAT alone improved significantly more than did those of the patients who received combined therapy”. This is quite remarkable, but I can´t find that this is being discussed later in the manuscript?

6.     The authors argue that the time frame for an optimal comparative study would be within the “golden period”, but timing of the final assessment should be also be discussed- in the study by Erbil  et al they assessed the patients at 3 months but the rest of the studies did their final assessment after treatment or at four weeks. Would it not be appropriate to investigate long-term effects, e.g. after six months, as well?
